# Monodisperse Micro-Droplet Generation in Microfluidic Channel with Asymmetric Cross-Sectional Shape

**DOI:** 10.3390/mi14010223

**Published:** 2023-01-15

**Authors:** Youngseo Cho, Jungwoo Kim, Jaewon Park, Hyun Soo Kim, Younghak Cho

**Affiliations:** 1Department of Mechanical System Design Engineering, Seoul National University of Science & Technology, 232 Gongneung-ro, Nowon-gu, Seoul 01811, Republic of Korea; 2OJEong Resilience Institute (OJERI), Korea University, 145 Anam-ro, Seongbuk-gu, Seoul 02841, Republic of Korea; 3Department of Electronic Engineering, Kwangwoon University, 20 Kwangwoon-ro, Nowon-gu, Seoul 01897, Republic of Korea

**Keywords:** monodisperse micro-droplet, asymmetric cross-section, micro-droplet generation, hypotenuse-to-width ratio (HTWR)

## Abstract

Micro-droplets are widely used in the fields of chemical and biological research, such as drug delivery, material synthesis, point-of-care diagnostics, and digital PCR. Droplet-based microfluidics has many advantages, such as small reagent consumption, fast reaction time, and independent control of each droplet. Therefore, various micro-droplet generation methods have been proposed, including T-junction breakup, capillary flow-focusing, planar flow-focusing, step emulsification, and high aspect (height-to-width) ratio confinement. In this study, we propose a microfluidic device for generating monodisperse micro-droplets, the microfluidic channel of which has an asymmetric cross-sectional shape and high hypotenuse-to-width ratio (HTWR). It was fabricated using basic MEMS processes, such as photolithography, anisotropic wet etching of Si, and polydimethylsiloxane (PDMS) molding. Due to the geometric similarity of a Si channel and a PDMS mold, both of which were created through the anisotropic etching process of a single crystal Si, the microfluidic channel with the asymmetric cross-sectional shape and high HTWR was easily realized. The effects of HTWR of channels on the size and uniformity of generated micro-droplets were investigated. The monodisperse micro-droplets were generated as the HTWR of the asymmetric channel was over 3.5. In addition, it was found that the flow direction of the oil solution (continuous phase) affected the size of micro-droplets due to the asymmetric channel structures. Two kinds of monodisperse droplets with different sizes were successfully generated for a wider range of flow rates using the asymmetric channel structure in the developed microfluidic device.

## 1. Introduction

Recently, micro-droplets have been widely used in various applications of biological and chemical research, such as drug delivery, material synthesis, point-of-care diagnostics, digital PCR, and single-cell analysis [1,2,3,4,5,6]. Droplet-based microfluidics has many advantages, such as small reagent consumption, fast reaction time, and independent control of each droplet. Therefore, various monodisperse micro-droplet generation methods have been proposed, including T-junction breakup [7,8,9], capillary flow-focusing [10], planar flow-focusing [11,12], step emulsification (SE) [13,14,15,16,17], and high-aspect-ratio (HAR) confinement [4,18] 

In the SE configuration, the interfacial tension between two immiscible phases by Laplace pressure difference drives micro-droplets to pinch off when passing through the stepped channel. Recently, the mechanisms of micro-droplet formation and various classical structures in step emulsification were reviewed [13]. The generation process in SE mainly depends on the interfacial tension, rather than a high-energy shear force, resulting in uniformly sized micro-droplets to be produced independent of the flow rates of both continuous and dispersed phases [19,20]. The SE method can provide stable and robust micro-droplet generation with a single size regardless of flow fluctuation. 

Recently, Yao et al. reported that a high-aspect-ratio (HAR) channel could generate monodisperse micro-droplets by interfacial surface tension [4]. Similar to the SE method, the HAR channel (aspect ratio > 3.5) created uniform micro-droplets without being affected by the flow rates. In addition, it enabled high-throughput micro-droplet generation by parallelizing the HAR channels on a single device. Previously, our group presented a microfluidic device with a HAR parallelogram cross-sectional channel, which allowed for more stable and robust generation of monodisperse micro-droplets at a wider range of flow rates [18]. Although both the SE and HAR methods are attractive for robust micro-droplet production, the biggest drawback is that only one size of micro-droplets can be obtained from a single device. The geometry of the microfluidic channel must be changed whenever micro-droplets with different sizes are required. To solve this problem, splitting of micro-droplets can be adapted for the simultaneous micro-droplet generation with different sizes [21,22,23].

In this study, a microfluidic channel comprising both an asymmetric cross-sectional shape and a high hypotenuse-to-width ratio (HTWR) was designed to generate micro-droplets of multiple sizes with high stability and uniformity. The device was fabricated using a simple and low-cost HAR channel fabrication method [18]. The effects of the HTWR on the size and uniformity of generated micro-droplets were investigated. The monodisperse micro-droplets were obtained when the HTWR of the asymmetric channel was over 3.5, and the asymmetric cross-sectional shape (trapezoidal shape) resulted in different sizes of micro-droplets to be produced depending on oil flow direction (continuous phase). Monodisperse micro-droplets with two different sizes were successfully generated for a wider range of flow rates using the developed microfluidic device. In addition to the experimental studies, we carried out simulations to further understand the micro-droplet breakup occurring in the asymmetrical channel geometry in a qualitative sense. 

## 2. Materials and Methods 

### 2.1. Design and Fabrication of the Microfluidic Device with an Asymmetric and Trapezoidal Cross-Sectional Channel 

Figure 1 shows the schematic of the developed microfluidic device consisting of a T-junction micro-droplet generator based on self-breakup mechanism. Here, the middle interconnecting channel in the T-junction region was designed to have an asymmetric and trapezoidal cross-sectional shape with a high HTWR (>3.5). When the sample solution passes through the middle HTWR channel, its micro-droplet breakup mechanism is dominated by the interfacial tension, similar to the self-breakup generation of HAR channel [4]. The width of the channel (W = W_2_ − W_1_) and the length of both hypotenuses (A: short hypotenuse, B: long hypotenuse) were determined by the photomask design; the master width (W_1_) for a PDMS mold and the Si channel width (W_2_). The width of channel (W) was fixed to 12 µm, and the other design parameters of the microfluidic channel with the asymmetric cross-sectional shape are shown in Figure 1b and Table 1.

Figure 2 illustrates the fabrication processes of the microfluidic channel with the asymmetric and trapezoidal cross-section, which was modified from the previously reported HAR channel fabrication method [18]. Briefly explained, (i) a (100) single crystal Si wafer with a SiO_2_ layer was patterned by photolithography and wet etching of buffered oxide etchant (BOE); (ii) the Si pattern was anisotropically etched in the potassium hydroxide (KOH) solution; (iii) the SiO_2_ layer was removed and additional wet etching of a Si channel was conducted after wafer dicing; (iv) a PDMS mold was replicated from a Si master; and (v) the Si channel and the PDMS mold were self-aligned and bonded by O_2_ plasma treatment. The Si channel substrate and the PDMS mold had the same crystal plane (111) since they were fabricated using anisotropic wet etching from the same Si wafer, and therefore, it was possible to align them easily based on their similar geometries.

### 2.2. Characterization of Micro-Droplet Generation 

Prior to the micro-droplet generation experiments, all channels of the fabricated microfluidic devices were coated with Aquapel^®^ (PGW LLC, Cranberry Township, PA, USA) to make their surfaces hydrophobic. It is known that the hydrophobic surface promotes the stable micro-droplet generation [24]. Then, the channels were rinsed with nitrogen gas and oil solution (FC-40 oil, 3M, Maplewood, NJ, USA). While the sample solution (disperse phase) flowed into the middle asymmetric channel, one of two sample inlets was clamped. The oil solution (continuous phase) was mixed with a surfactant (008-FluoroSurfactant, RAN Biotechnologies, Beverly, MA, USA) at 2% (w/w) ratio for stable micro-droplet generation and storage. Both the sample and oil solutions were injected using syringe pumps (LEGATO 111, KD Scientific Inc., Holliston, MA, USA). Micro-droplet generation was monitored using an optical microscope (BX-60, Olympus, Tokyo, Japan) equipped with a high-speed camera (VEO E310L, Phantom, Wayne, NJ, USA). The generated micro-droplets were collected through the outlet, and their sizes were measured with bright-field microscopy and a ImageJ software. The data shown in this research were analyzed from at least 30 different micro-droplets.

To investigate how the HTWR and the sample flow rate affects the micro-droplet generation, microfluidic devices with three different interconnecting channel designs (Channel_A, Channel_B, Channel_C) were tested and compared, where all three designs had the same channel width (W = 12 µm) but different lengths of hypotenuses (Table 1). The flow rate of the oil solution was fixed at 200 µL/h, and micro-droplet generation under various flow rate conditions of the sample solution (30, 50, 100, 150, and 200 µL/h) were analyzed. The developed asymmetric and trapezoidal cross-sectional channels have different lengths of hypotenuses, so the sample solution experiences different flow profiles at the T-junction region depending on the oil flow direction. To examine the effect of the different flow profiles at the interface, micro-droplet generation was compared using different oil flow direction. In addition, the flow rates of 100, 200, and 400 µL/h in the oil solution were tested to examine their influence on the produced micro-droplet size. The flow rate of the sample solution was maintained at 50 µL/h. 

### 2.3. Simulation of Micro-Droplet Generation in Different Channel Types 

To obtain additional understanding of the micro-droplet breakup occurring in the asymmetrical channel geometry, numerical simulations were performed using a commercial software package, ANSYS Fluent. The volume of fluid model was employed to track the volume fraction of the sample solution (i.e., micro-droplet) and oil solution. The methods considered here are similar to those of Ji et al. [18]. In the simulations, the channels considered were W12-A45-B54 and W12-A36-B45, which corresponds to the Channel_A and Channel_B designs, respectively. For the computational efficiency, a simplified T-junction (would be shown later) is considered for the purpose of understanding the micro-droplet generation phenomenon rather than reproducing the experimental device. Information related to micro-droplet breakup, such as void fraction and micro-droplet breakup position, was analyzed. The number of grid points was approximately 1.1 million. This grid resolution is at the level of considering about 10 cells for the diameter of one bubble, which is usually observed in the literature. As boundary conditions, the uniform velocity conditions were used in the main channel (oil flow channel) and branch channel (HTWR asymmetric channel). Additionally, the pressure conditions were adopted in the outlet of the channel.

## 3. Results and Discussion

### 3.1. Microfluidic Channel with an Asymmetric Cross-Section and High Hypotenuse-to-Width Ratio 

Figure 3a depicts the fabricated microfluidic device with the asymmetric channel, which consists of two inlets for sample solution and oil solution, an outlet for collecting the generated micro-droplets, and an interconnecting trapezoidal channel for a T-junction micro-droplet generation. Figure 3b-d show the scanning electron microscope (SEM) images of asymmetrical channel cross-sections, where different lengths of left and right hypotenuses of each channel design were clearly observed. For example, in Figure 3b, both sides (A: short hypotenuse, B: long hypotenuse) of the fabricated channel have the HTWR larger than 3.5 (W: the channel width). In Figure 3c,d, the channels are characterized by different values of HTWR depending on the sides, where short hypotenuse (A) has the HTWR smaller than 3.5, while another side (B: long hypotenuse) displays the HTWR larger than 3.5. The cross-sectional shapes of each channel could be easily controlled, where the width (W) and hypotenuses (A and B) of the fabricated channels were determined by the width difference of photomask design (W_1_ and W_2_).

### 3.2. Characterization of Micro-Droplet Generation with Different Channel Geometries 

The time-lapse images of the micro-droplet generation process at the water–oil interface on both oil flow direction (A→B and B→A) are illustrated in Figure 4. As the sample solution approached the T-junction region, the sample solution was pressurized, and a thin thread pinched off. Once the micro-droplet was formed and released to an oil-flowing channel, the elongated thread retracted and returned to the initial state. These steps were repeated and resulted in monodisperse micro-droplets produced periodically (Appendix A).

The effects of the sample flow rate as well as the oil flow direction on the size of generated micro-droplets were investigated using three different asymmetric channel designs (Channel_A, Channel_B, and Channel_C). The three different channel designs have all trapezoidal shapes with the same channel width of 12 μm, but with different hypotenuse lengths on both sides, providing the information on how the HTWR affects the micro-droplet generation.

Figure 5 summarizes the microscopic images of generated micro-droplets and the analysis of their diameters in the three different channel designs under various sample flow rate and oil flow direction conditions. Uniformly sized micro-droplets were generated when the oil solution flowed from A (short hypotenuse) to B (long hypotenuse), where the HTWR (B/W) of the all three channel designs was higher than 3.5. However, when the oil flow changed in opposite direction (B→A), particularly if the HTWR (A/W) was less than 3.5 (Channel_B and Channel_C), the micro-droplet sizes changed depending on the sample flow rates, as shown in Figure 5c,d.

In the Channel_A design, where the HTWR of both sides in the asymmetric channel was higher than 3.5 (A/W > 3.5, B/W > 3.5), monodisperse micro-droplets with diameters of 51.8 ± 0.9 and 61.9 ± 1.2 μm were generated in the A→B and B→A directions, respectively, for all tested sample flow rates (ranging from 30 to 200 μL/h, Figure 5b). In other words, the variation of the sample flow rates did not affect the micro-droplet generation, clearly demonstrating that uniformly sized micro-droplets were generated independent of sample flow rates. This result is consistent with the previously reported study [4]. One interesting feature here is that different sizes of micro-droplets were produced simply by changing the oil flow direction. Within the same oil flow direction, the generated micro-droplet size was kept constant, but when the flow direction changed in opposite way, different size of monodisperse micro-droplets was produced, where their size difference is about 10 µm. The size variation according to the oil flow direction will be further discussed in next section with simulation results. The average sizes of the micro-droplets were kept almost consistent, where less than 1.9% size variation was observed over the wide range of sample flow rates from 30 to 200 μL/h. 

In the Channel_B and Channel_C designs (both A/W < 3.5, B/W > 3.5), monodisperse micro-droplets were produced in the A→B directions, similar to the Channel_A design (Figure 5c,d). However, when the channel structure with the HTWR of less than 3.5 was used in micro-droplet generation, the micro-droplet breakup profile followed a conventional T-junction micro-droplet generation principle, which was governed by the viscous shearing forces between oil and sample solutions. In other words, in the B→A direction, the size of the produced micro-droplet is strongly dependent on flow variations, and micro-droplets with larger diameters were formed as the higher sample flow rates were applied. For example, in the Channel_B design, the average diameter of the micro-droplets was 51.1 (A→B direction) and 38.2 µm (B→A direction) under the sample flow rate condition of 30 µL/h. As the sample flow rate became higher, larger micro-droplets were generated in the B→A direction (at 200 µL/h, average diameter: 208.2 µm) while the micro-droplet size was kept consistent in the A→B direction. 

As illustrated in Figure 6, micro-droplet generation experiments to examine the effect of oil solution were conducted by changing the oil flow rates from 100 to 400 μL/h and the oil flow direction. The sample flow rate was fixed at 50 μL/h during the experiment, and the micro-droplet formation was monitored. In the Channel_A design, as mentioned in the above, monodisperse but different-sized micro-droplets were generated regardless of the oil flow rates and direction (Figure 6b). In the Channel_B and Channel_C designs, uniformly sized micro-droplets were produced in the A→B directions independent of oil flow rates (B/W > 3.5). However, the diameter of the produced micro-droplets is strongly dependent on flow variations for the B→A direction (A/W < 3.5), where micro-droplet size decreased as the oil flow rate increased (Figure 6c,d). Moreover, the diameter of micro-droplets from Channel_B was smaller than that from Channel_C, indicating that the micro-droplet breakup profile followed a conventional T-junction micro-droplet generation principle in the B→A direction (HTWR < 3.5). From all experiment results, it is clearly confirmed that the developed microfluidic device not only generated monodisperse micro-droplets independent of sample and oil flow rates, but also created two different size micro-droplets on a single platform by controlling the oil flow direction. 

### 3.3. Simulation Results 

Figure 7 and Figure 8 show the void fraction of the sample solution in the two different asymmetric channel designs, W12-A36-B45 and W12-A45-B54, considered in this study. In the simulations, the flow rate of the continuous oil solution was fixed at 200 μL/h while changing the flow rate of the dispersed sample solution with 50 and 100 μL/h. The effect of different oil flow direction was also analyzed as conducted in the experiments.

The simulation results indicate that the micro-droplet breakup mechanism depends on the HTWR (A/W or B/W) as well as the oil flow direction in the asymmetrical channel. When the HTWR (A/W or B/W) is larger than 3.5, the micro-droplet breakup usually occurs inside the middle interconnecting channel (see Figure 7a and Figure 8a,b). On the other hand, as shown in Figure 7b, when the HTWR is smaller than 3.5, the micro-droplet breakup exhibits different behavior, where the shear force is dominant, and the micro-droplet generation follows the conventional T-junction breakup mechanism. In Figure 7, at the A→B direction (HTWR > 3.5), the sample solution flowing in the interconnecting channel is concentrated in the center or corner (see Figure 7a). Therefore, at the A→B direction, the necking of the sample solution is found within the interconnecting channel. On the other hand, at the B→A direction (HTWR < 3.5), the micro-droplet breakup is also observed in the center or corner as explained in the above for the low sample flow rate. However, the sample solution fills more the interconnecting channel as the oil solution flow rate increases (see Figure 7b). As a result, the different breakup of the micro-droplet with respect to the oil solution flow rate could result in the increase in the droplet size shown in Figure 5, which is consistent with the previously reported study [18]. 

## 4. Conclusions

In this paper, we developed the microfluidic device comprising the high hypotenuse-to-width ratio (HTWR) channel, which allowed for monodisperse micro-droplet generation. The HTWR channel has the asymmetric and trapezoidal cross-section, where the size of the produced micro-droplet can be easily adjusted by changing the oil flow direction. In contrast to previously developed HAR and SE channel structures, the developed asymmetric channel design was able to produce two different sizes of monodisperse micro-droplets from a single device. The developed device was fabricated using simple and cheap MEMS processes, where the asymmetric channel geometry was successfully created and integrated into the microfluidic device using self-alignment between the Si substrate and PDMS mold. From the device characterization under various sample flow rate conditions, the developed device successfully produced uniformly sized micro-droplets regardless of the flow fluctuations. The effect of the oil flow rate and direction on the micro-droplet generation was also examined, and monodisperse microdroplets with two different sizes were successfully generated, only dependent on flow direction. The simulation results showed the micro-droplet breakup mechanisms in the asymmetric channel and how two different hypotenuse sides affected the micro-droplet generation. These results clearly demonstrate the capability of the developed device: generation of two different sizes of uniform micro-droplets on a single device.

## Figures and Tables

**Figure 1 micromachines-14-00223-f001:**
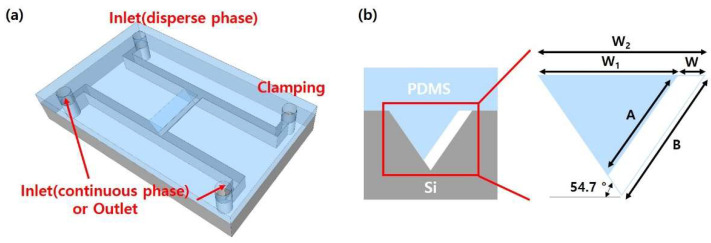
(**a**) Schematic view of the developed microfluidic device comprising a middle interconnecting channel with an asymmetric cross-section in the T-junction region, (**b**) Design parameters for the HTWR trapezoidal channel with the asymmetric cross-section.

**Figure 2 micromachines-14-00223-f002:**
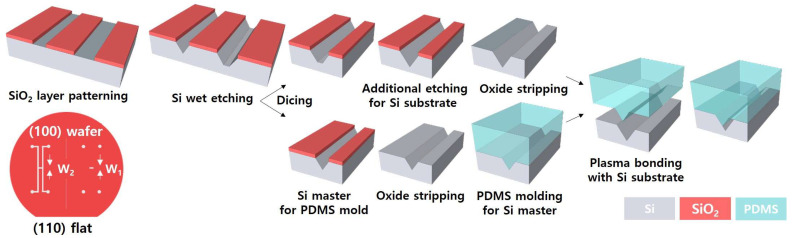
Fabrication process of the microfluidic devices with the asymmetric channel.

**Figure 3 micromachines-14-00223-f003:**
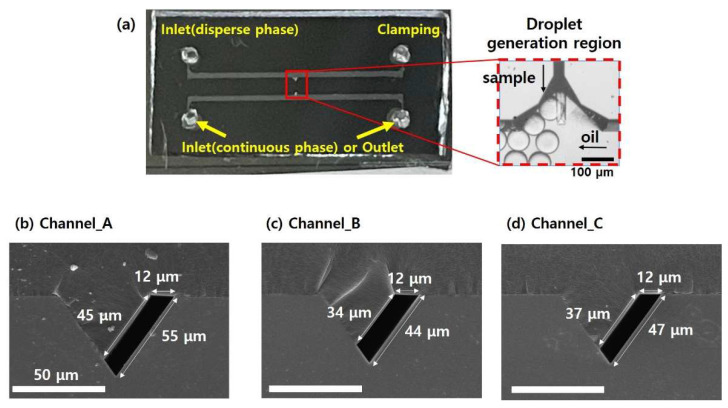
(**a**) Photographs of the developed microfluidic device consisting of the asymmetric cross-sectional channel. The inset illustrates the close-up view of the T-junction micro-droplet generation region. (**b**–**d**) The SEM images showing the cross-section of the fabricated channels. (**b**) Cross-section of Channel_A with A/W > 3.5 and B/W > 3.5, (**c**) Cross-section of Channel_B with A/W < 3.5 and B/W > 3.5, (**d**) Cross-section of Channel_C with A/W < 3.5 and B/W > 3.5.

**Figure 4 micromachines-14-00223-f004:**
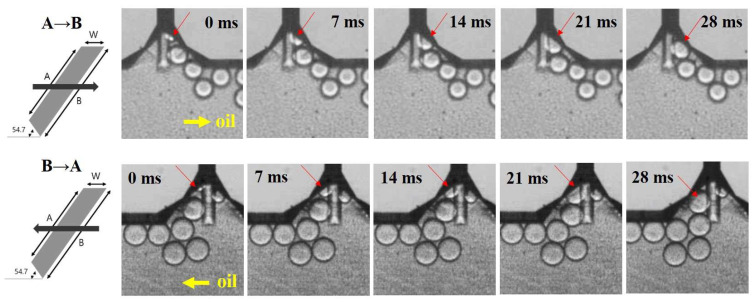
Time-lapse images displaying the micro-droplet generation process at the T-junction region with different oil flow direction (Top: A→B, Bottom: B→A, Channel_A design, oil flow rate: 200 μL/h, water (sample) flow rate: 50 μL/h).

**Figure 5 micromachines-14-00223-f005:**
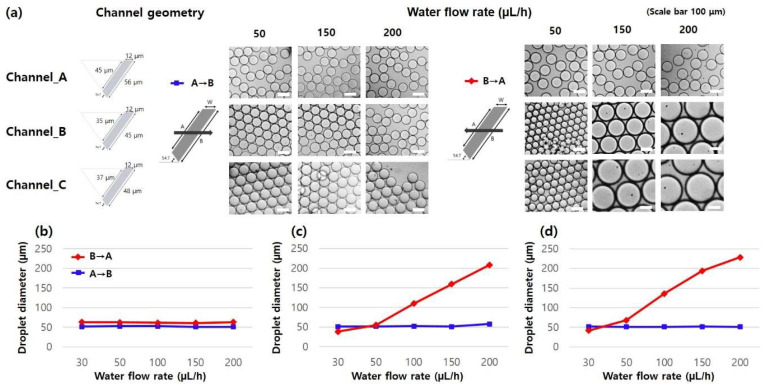
(**a**) Microscopic images of micro-droplets generated under various sample flow rate and oil flow direction conditions. Average diameters of micro-droplets generated in (**b**) Channel_A, (**c**) Channel_B, and (**d**) Channel_C designs. The oil flow rate was fixed at 200 μL/h.

**Figure 6 micromachines-14-00223-f006:**
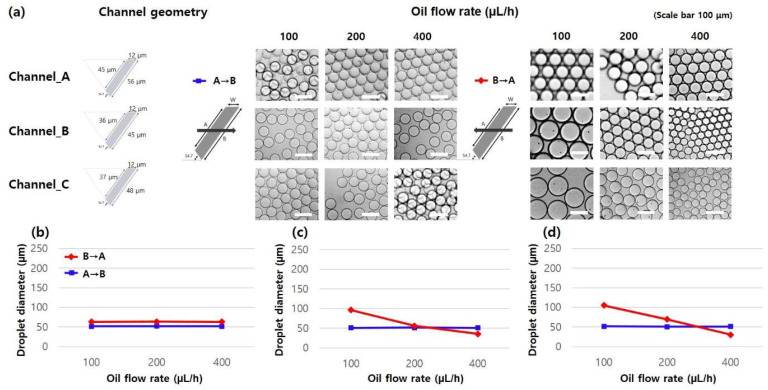
(**a**) Microscopic images of micro-droplets generated under various oil flow rate and direction conditions. Average diameters of micro-droplets generated in (**b**) Channel_A, (**c**) Channel_B, and (**d**) Channel_C designs. The flow rate of the water solution was fixed at 50 μL/h.

**Figure 7 micromachines-14-00223-f007:**
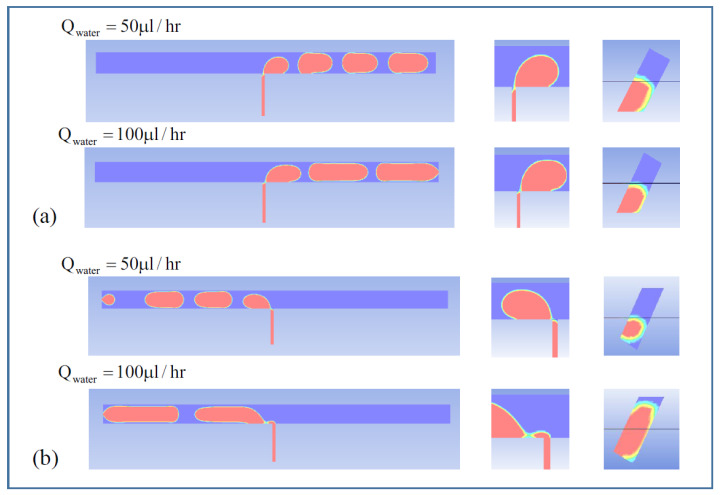
Void fraction of micro-droplet phase in the channel of W12-A36-B45. (**a**) A→B direction, (**b**) B→A direction.

**Figure 8 micromachines-14-00223-f008:**
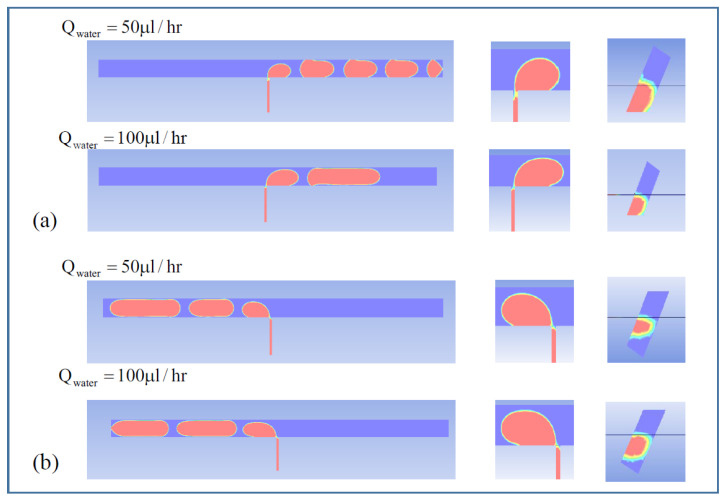
Void fraction of micro-droplet phase in the channel of W12-A45-B54. (**a**) A→B direction, (**b**) B→A direction.

**Table 1 micromachines-14-00223-t001:** Design parameters of the microfluidic channels with asymmetric cross-sections.

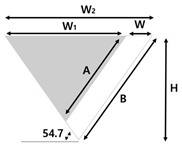	**Sample**	**W_1_**	**W_2_**	**W**	**H**	**A**	**B**	**A/W**	**B/W**	**H/W**
	Channel_A	64	52	12	45	45	56	3.8	4.6	3.7
	Channel_B	52	40	12	36	35	45	2.9	3.8	3.0
	Channel_C	55	43	12	39	37	48	3.1	4.0	3.3

## Data Availability

Not applicable.

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
