# Peer review of "Monodisperse Micro-Droplet Generation in Microfluidic Channel with Asymmetric Cross-Sectional Shape"

_micromachines, 2023, doi:10.3390/mi14010223_

Round 1

Reviewer 1 Report

A microfluidic device with a trapezoidal cross-section (of appropriate HTWR) in the T-junction region is proposed for the generation of monodisperse droplets of two sizes depending on the flow direction of the continuous phase (oil solution). The manuscript is well-written and the work well-described. Minor improvements are recommended.

-        Figure 1a should show clearly and in agreement with Figure 3a that the inlet (oil) and outlet (oil +droplet) are used interchangeably for adjusting the oil flow direction.

-        Simulation for micro-droplet generation: a grid of 1.1 million points was implemented. Did the authors check the number of grid points ensuring solution independency?

-        Line 258: why the channel designs used in the simulation were not exactly the same as those used in the experimental work (Table 1)?         

Reviewer 2 Report

In this paper, the authors proposed a microfluidic device for generating monodisperse micro-droplets, the microfluidic channel of which has an asymmetric cross-sectional shape and high hypotenuse-to-width ratio (HTWR). Two kinds of monodisperse droplets with different sizes were successfully generated for a wider range of flow rates using the asymmetric channel structure in the developed microfluidic device. They used simulation software to simulate generation of droplets and charactered micro-droplet generation with different channel geometries, the results are convincing enough. Toward improve the manuscript, the authors are suggested to consider the following comments.

1. There is a problem with expression in English, the writing is lengthy and sometimes hard to follow and words are relatively simple. It is suggested to organize the manuscript in a concise way.

2. It makes more sense to put the simulation results before the experimental results.

3. The icon position should be consistent, all in the upper left corner or in the middle of the pictures (Figure 5 and Figure 6).
